REGISTERED REPORT PROTOCOL

# Measurement properties of pain scoring instruments in farm animals: A systematic review protocol using the COSMIN checklist

**Rubia Mitalli Tomacheuski[1], Beatriz Paglerani Monteiro[2], Marina Cayetano Evangelista[2], Stelio Pacca Loureiro Luna[3], Paulo Vinícius Steagall**[1,2]*

**1** Department of Anesthesiology, Medical School (FMB) of São Paulo State University (UNESP), Botucatu, São Paulo, Brazil, **2** Département de sciences cliniques, Faculté de médecine vétérinaire, Université de Montréal, Saint-Hyacinthe, Québec, Canada, **3** Department Veterinary Surgery and Animal Reproduction, School of Veterinary Medicine and Animal Science, São Paulo State University (UNESP), Botucatu, São Paulo, Brazil

* paulo.steagall@umontreal.ca

## Abstract

Society has been increasingly concerned about the impact of pain on farm animal welfare. This systematic review aims to provide evidence relating to the measurement properties (i.e. reliability, validity, and sensitivity) of pain scoring instruments used for pain assessment in farm animals. A literature search will be performed using five databases (MEDLINE, EMBASE, Web of Science, CAB abstracts and Biological Abstracts) and search terms related to pain, pain scales and different species of farm animals. Eligibility criteria will include full-text studies on the development and/or validation of acute and chronic pain scoring instruments for farm animals including bovine (beef and dairy), ovine, caprine, camel, swine and poultry. Exclusion criteria will include studies that report the use of pain scales for the validation of another instrument, or those reporting ethograms/list of behaviors potentially indicating pain without a scoring system. Study titles and their abstracts will be screened for eligibility by one investigator. Full-text articles will be independently reviewed for eligibility and evaluated by two investigators. Relevant information will be recorded and evaluated systematically according to the Consensus-based Standards for the Selection of Health Measurement Instruments (COSMIN) checklist using an adapted data collection sheet. The following measurement properties and characteristics of the instruments will be assessed: content validity (internal consistency, structural and cross-cultural validity), reliability, measurement error, criterion and construct validity, responsiveness, interpretability and feasibility. Following the assessment of methodological quality and quality of the findings, evidence for each measurement property will be summarized into high, moderate, low or very low. This systematic review will provide further insights into the evidence-based measurement properties of pain scoring instruments in farm animals. It may identify possible gaps of knowledge with these tools as a potential target for future studies in farm animals with a positive impact on animal welfare.

**Data Availability Statement:** All relevant data from this study will be made available upon study completion.

**Funding:** The work of RT was funded by a scholarship from Coordenação de Aperfeiçoamento de Pessoal de Nível Superior (CAPES), Brazil (001) and a grant from The São Paulo Research Foundation (FAPESP) Thematic Projects (2017/12815-0).

**Competing interests:** The authors have declared that no competing interests exist.

# Introduction

Society has been increasingly concerned about the impact of pain on farm animal welfare [1]. Farm animals are less frequently treated for pain when compared with companion animals [2] and horses [3]. Possible reasons for this include the misconception that farm animals do not feel as much pain as small animals, concerns related to withdrawal times of analgesics for human food safety, lack of knowledge about assessing pain in farm animal species [3, 4], and budget considerations for the cost of analgesic therapies combined with the low zootechnical and affective value of farm animals [5–8]. Pain causes suffering, fear and stress, negatively impacting animal welfare and sometimes decreasing productivity [5, 9, 10]. Pain recognition and measurement are important components of animal welfare [5].

Pain assessment in animals is commonly performed through evaluation of species-specific behaviors [11] and changes in facial expressions [12–14]. Other methods of pain assessment include the use of quantitative sensory testing for evaluation of the animals' sensory profile [15] and the use of kinetics or kinematics for evaluation of levels of activity and lameness [16–18]. However, these outcome measures require specific equipment and training and are not readily available in practice. Surrogate measures of pain might also include animal production outcomes, physiological parameters, and biomarkers [19–21]; yet these are not necessarily specific to pain. For these reasons, in practice, pain assessment relies on the evaluation of behaviors that could be associated with pain (including facial expressions) using scoring instruments (i.e. scales, tools, metrology instruments, etc.). Pain scoring instruments are non-invasive, inexpensive, do not require any equipment or restraint and may be performed by remote observation [22]. They are used to identify and quantify pain, and to monitor response to analgesic treatments. These instruments focus on the behavioral expression of pain and generally include a systematic description of behaviors accompanied by their respective scores. When such behaviors only involve facial expressions, they are known by facial expression scales or grimace scales. Pain scoring instruments have been developed for farm animals and may include assessment of activity, body posture, response to interaction, attention to wound/painful area, and/or facial expressions [14, 22–26]. In ruminants, for example, the most frequently observed behaviors that are possibly associated with pain include changes in appearance, posture, gait, appetite, interaction with other animals and the environment, decreased or increased frequency of locomotion, weight bearing, vocalization, increased attention to the injured area, lip-licking, curved lips, teeth grinding, tremors and strong tail wagging [5, 26–29]. Similarly, behaviours that could be associated with pain and changes in facial expressions have been identified in swine [14, 22]. In poultry, there is a lack of studies regarding pain assessment; however, change or absence of normal behaviours have been described including decreased social interactions, increased aggression, showing guarding and/or grooming behaviour [30]. Unidimensional scales such as the numerical rating scale (NRS), simple descriptive scale (SDS) and visual analog scales (VAS) have been used in the past to measure postoperative pain in sheep [31, 32]. However, these tools are not considered adequate because they were developed and validated for humans who self-report their degree of pain; these scales are subjective, not species-specific and influenced by the level of familiarity/expertise of the observer [26, 33, 34]. Species-specific pain scales have been developed for use in farm animals, such as sheep, cattle and pigs, and different levels of validation have been reported for some of these instruments [14, 22, 23, 26, 35–37]. Nevertheless, there is lack of validated instruments for some species of farm animals, like goats, camels and poultry.

Pain scoring instruments need to undergo several steps of scientific validation to ensure they are valid and reliable before they can be used in practice with confidence. In order to evaluate whether an instrument is valid and reliable, one must assess the measurement (or

psychometric) properties of such instrument. Measurement properties refer to the characteristics or attributes of an instrument which are a consequence of the methodology used in their respective studies. In other words, measurement properties refer to the quality of the methodology. The following are the most commonly reported measurement properties of pain scoring instruments: a) development/content validity (expert assessment of the items included in the scale, the calculation of a content validation index, development of ethograms and/or evidence from the literature [38, 39]); b) structural and/or cross-cultural validity [39–41]; c) internal consistency (degree of the interrelatedness among the items [39, 41]); d) measurement error (systematic and random error in a patient's score that is not associated to real changes in the construct to be assessed including sensitivity, specificity and accuracy [41]); e) reliability (whether the scores are consistent between different observers and over time, known as inter- and intra-observer reliability, respectively [22, 39]); f) criterion validity (correlation of the proposed tool with other existent scales [39, 41]); g) construct validity (whether the tool measures what it is supposed to measure, in this case, pain, by comparing different known groups and if the scores decrease after rescue analgesia [39, 41]); h) responsiveness (ability to detect changes over time); and i) a definition of a cut off point for administration of rescue analgesia [22, 38, 42].

Systematic reviews of outcome measurement instruments (e.g. pain scoring instruments) are important for selecting the most suitable instrument to measure a construct of interest (i.e. pain) in the target study population [43]. To the authors' knowledge, systematic reviews on the evidence of the measurement properties of different pain scoring systems in farm animals have not been published.

## Objective

This systematic review aims to provide evidence relating to the measurement properties (i.e. reliability, validity and sensitivity) of pain scoring instruments used for pain assessment in farm animals using the Consensus Based Standards for the Selection of Health Measurement Instrument (COSMIN) methodology [41, 44, 45].

## Materials and methods

### Databases and search terms

Five bibliographic databases (MEDLINE via PubMed, EMBASE, Web of Science, and CAB abstracts and Biological Abstracts via Web of Science) will be searched to identify studies published in peer-review journals. There will be no publication period nor language restriction. The search terms were defined using MeSH (Medical Subject Headings), a controlled vocabulary thesaurus produced by the National Library of Medicine, which index articles for MEDLINE/PubMed and using DeCS (Health Science Descriptors), a structured and multilingual vocabulary used as a unique language in indexing articles from scientific literature via the Virtual Health Library, which includes databases such as LILACS, MEDLINE, PAHO IRIS Repository, BIGG International database GRADE guidelines, BRISA Regional Base of Health Technology Assessment Reports of the Americas, CARPHA EvIDeNCe Portal, Observatorio Regional de Humanos de Salud, and PIE Evidence-Informed Policies.

The chosen search terms have been refined and tested using PubMed. The following descriptor items will be included: ("pain scoring system*" OR "pain scale*" OR "pain indicator*" OR "grimace scale*" OR "facial expression*" OR "pain behavior*" OR "pain assessment*") AND ("farm animal*" OR ruminant* OR bovine OR beef OR cattle OR cow OR cows OR buffalo* OR camel* OR ovine OR sheep* OR lamb* OR goat* OR caprine* OR

swine OR porcine OR pig OR pigs OR piglet* OR poultry* OR chicken* OR fowl* OR duck* OR geese).

## Eligibility criteria

Original studies reporting the development and/or validation of pain scoring instruments for the assessment of pain in farm animals as well as manuscripts reporting the assessment of one or more measurement properties of these instruments, will be included. These studies may involve naturally-occurring or experimental acute and chronic painful conditions in bovine (beef and dairy, including buffalo), ovine (sheep and lamb), caprine (goat and kid), camel, swine (pig and piglets) and poultry (chicken, fowl, ducks, turkeys, and geese). These species were chosen since they are the most relevant species used for production of animal protein (meat, dairy products and eggs) according to the latest report from the Organisation for Economic Co-operation and Development (OECD) and the Food and Agriculture Organization (FAO) of the United Nations, the OECD-FAO Agricultural Outlook 2020–2029 [46].

Studies that only report the use of pain scales as an outcome measurement instrument (e.g. in randomized controlled trials comparing two different treatments), studies in which a pain scale is used in the validation of another instrument, studies reporting only ethograms/list of behaviors that could be indicators of pain without a scoring system, studies reporting non-ordinal pain assessment variables, or review and systematic reviews will not be included. Studies reporting the use of pain scoring instruments to measure constructs other than pain, for example studies assessing animal welfare, in which pain is considered within the overall evaluation, studies assessing nociceptive testing, and studies for which the full text is not available will be excluded.

## Literature search

Study titles and their abstracts will be initially screened for eligibility by one investigator (RMT) using the search strategy described above. Full-text articles will be selected. The references will be exported into Endnote (version X9) and Covidence (a web-based software platform integrated with the Cochrane's review production toolkit that streamlines the production of systematic reviews), and duplicates will be removed.

Full-text articles will be independently reviewed for eligibility criteria by two investigators (RMT and BPM) using Covidence. "Snowball" methods such as pursuing references of eligible articles and/or reviews and electronic citation tracking will be used to maximize the retrieval of relevant studies.

## Data extraction

Data from included studies will be extracted by one reviewer (RMT) using a predefined data collection sheet (excel format). The following information will be extracted:

1. characteristics of the study population (age; gender; breed/strain; where/how animals were housed; how animals were handled; duration and source of pain);

2. characteristics of the scale (name/version; language/translation; scoring method; number and name of items/action units);

3. setting and purpose for which the scale is intended (e.g. chronic or acute pain; adult or juvenile/pediatric animals; hospital, experimental or commercial setting), interpretability, and operational characteristics such as the feasibility of administration for users (i.e. time required for completion of the instrument; who are the end-users; whether training is

required; whether evaluations can be done in real-time, or using image or video assessment).

## Assessment of the measurement properties

The quality assessment and summary of evidence will be performed independently by two reviewers (RMT and BPM) using an excel file. All information will be recorded and evaluated systematically as adapted from the COSMIN (COnsensus-based Standards for the selection of health Measurement INstruments) checklist [41]. The COSMIN is an initiative of an international multidisciplinary team of researchers with a background in epidemiology, psychometrics, qualitative research, and health care, who have expertise in the development and evaluation of outcome measurement instruments. It aims to improve the selection of outcome measurement instruments in research and clinical practice [41]. The COSMIN methodology was specifically developed and validated for use in reviews of patient-reported outcome measures [41, 44, 45, 47]. However, it can be adapted and used for other types of outcome measurement instruments such as those in which pain is not self-reported and is evaluated by proxy, which is the case in veterinary medicine [45]. For these reasons, an adapted COSMIN evaluation sheet will be used in which items such as comprehensibility (by the patient point of view) and methods of interviewing will not be assessed. The following measurement properties will be evaluated: content validity (including internal consistency, structural and cross-cultural validity), reliability, measurement error, criterion and construct validity, and responsiveness. Moreover, interpretability and feasibility will be evaluated. If the reviewers (RMT and BPM) are unable to reach a consensus on the assessment of measurement properties, a third reviewer will be consulted (PVS). Table 1 shows the general overview of the adapted COSMIN evaluation sheet.

First, each study will be assessed for methodological quality, rated as: very good, adequate, doubtful, inadequate or not applicable; and quality of the findings, rated as: sufficient or positive [+], when the majority of the summarized results meet the criteria for good measurement properties; insufficient or negative [−], when the majority of the summarized results do not meet the criteria for good measurement properties; conflicting findings [+/-] or indeterminate [?]. Thereafter, the evidence for each measurement property will be summarized into high, moderate, low or very low for each pain scoring system.

Reporting of the findings of this systematic review will be done according to the recommendations from PRISMA (Preferred Reporting Items for Systematic Reviews and Meta-Analyses) and the 10-step guideline from COSMIN.

## Impact

Pain assessment and management are fundamental parts of animal welfare. This systematic review will provide further insights into the evidence-based measurement properties of pain scoring instruments in farm animals. It may identify gaps in knowledge with these tools as a potential target for future studies in farm animals to improve in animal welfare.

## Limitations

Evidence might be limited or unavailable for some measurement properties or some species. As previously described, the COSMIN checklist was developed for assessing patient-reported outcome measures. In the case of non-verbal animals, pain assessment relies on evaluation by proxy using direct observation of behaviors or facial expressions using pain scales and/or grimace scales. For this reason, some items (i.e. comprehensibility, methods of interviewing) will not be assessed.

**Table 1. General overview of the criteria used for assessment of methodological quality.**

| Components of scale validation | Categories | Considerations |
|---|---|---|
| Scale development | General design requirements and development | Clearness of the description provided for the construct (origin, conceptual framework, disease model). Context in which the scale was developed including target population and sampling. Methods used to identify and define the items or action units. Handling of animals during evaluation. Potential biases. |
| | Content validity | If and how content validity was assessed and reported. |
| | Comprehensibility | If and how comprehensibility was evaluated (e.g. if the items and response options were adequately worded and understood by evaluators). |
| Measurement properties | Internal consistency | If and how internal consistency was calculated and reported, and if there are any flaws. |
| | Reliability | If and how of inter and intra-rater reliability were assessed and reported (e.g. number of raters, interval used for the reliability testing, statistical analysis and if there are any flaws). |
| | Measurement error | If and how sensitivity, specificity and/or accuracy were determined and if there are any flaws. |
| | Criterion validity | If and how criterion validity was assessed and reported (e.g. which instrument was used for comparison, its adequacy, appropriateness of the statistical methods used) and if there are any other flaws. |
| | Construct validity | If and how construct validity was assessed and reported (e.g. description of important characteristics of the subgroups, appropriateness of statistical methods used for the hypothesis(es) to be tested) and if there are any other flaws. Construct validity will be considered as discrimination between painful and pain-free animals. |
| | Responsiveness | If and how responsiveness was assessed and reported (e.g. clear description of the intervention given, appropriateness of statistical method used for the hypotheses to be tested) and if there are any other flaws. Responsiveness will be considered as discrimination between before and after analgesic intervention. |
| | Cross-cultural validity | If and how the instrument was adequately translated and validated in other languages (e.g. if translation and back translation were performed, if the samples were similar for relevant characteristics) and if there are any other flaws |

Adapted from the Consensus-based Standards for the Selection of Health Measurement Instruments (COSMIN) [41, 45, 48].

## Supporting information

**S1 File.**
(DOCX)

## Acknowledgments

Ms. Marie-Claude Poirier for the invaluable help with databases, search terms and literature search.

## Author Contributions

**Conceptualization:** Paulo Vinícius Steagall.

**Investigation:** Rubia Mitalli Tomacheuski, Beatriz Paglerani Monteiro, Marina Cayetano Evangelista, Paulo Vinícius Steagall.

**Methodology:** Rubia Mitalli Tomacheuski, Beatriz Paglerani Monteiro, Marina Cayetano Evangelista, Stelio Pacca Loureiro Luna, Paulo Vinícius Steagall.

**Project administration:** Beatriz Paglerani Monteiro, Paulo Vinícius Steagall.

**Resources:** Marina Cayetano Evangelista, Stelio Pacca Loureiro Luna, Paulo Vinícius Steagall.

**Supervision:** Beatriz Paglerani Monteiro, Paulo Vinícius Steagall.

**Validation:** Rubia Mitalli Tomacheuski, Beatriz Paglerani Monteiro, Marina Cayetano Evangelista.

**Writing – original draft:** Rubia Mitalli Tomacheuski.

**Writing – review & editing:** Beatriz Paglerani Monteiro, Marina Cayetano Evangelista, Stelio Pacca Loureiro Luna, Paulo Vinícius Steagall.

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
