## [Decision Letter · Decision Letter 0]

4 Feb 2021

PONE-D-20-37059

Measurement properties of pain scoring systems in farm animals: a systematic review protocol using the COSMIN checklist

PLOS ONE

Dear Dr. Steagall,

Thank you for submitting your manuscript to PLOS ONE. After careful consideration, we feel that it has merit but does not fully meet PLOS ONE’s publication criteria as it currently stands. Therefore, we invite you to submit a revised version of the manuscript that addresses the points raised during the review process.

Manuscript lacks in the quality of preparation. The major flaw of the study is the interpretation of the data, methodology, improving the clarity of arguments, English language and redaction style. Please revise all comments with your logical answers to these criticisms in terms of improving the manuscript. I agree with reviewers, and authors should improve the manuscript. Please review the referee comments and make your peer revision.

We look forward to receiving your revised manuscript.

Kind regards,

Arda Yildirim, Ph.D.

Academic Editor

PLOS ONE

Journal Requirements:

2. We note that some of our criteria for systematic reviews have not been met. We require that the quality of studies is assessed (and the results of the assessment provided as a supplemental file). More information on the requirement for quality assessment can be found in the PRISMA Elaboration and Explanation supplementary reference (http://prisma-statement.org/PRISMAStatement/PRISMAEandE). Thank you for your attention to our requests.

Additional Editor Comments:

This manuscript is interesting; however there is a major flaw in the interpretation of the data, methodology, improving the clarity of arguments, English language and redaction style. It is necessary to improve the manuscript by examining the questions that need to be clarified in a way. Please check your financial disclosure. For your guidance, you can check the reviewers' comments. Thank you for giving us the opportunity to consider your work.

Reviewers' comments:

Reviewer's Responses to Questions

**Comments to the Author**

1. Does the manuscript provide a valid rationale for the proposed study, with clearly identified and justified research questions?

Reviewer #1: Yes

Reviewer #2: Yes

Reviewer #3: Partly

2. Is the protocol technically sound and planned in a manner that will lead to a meaningful outcome and allow testing the stated hypotheses?

Reviewer #1: Yes

Reviewer #2: Yes

Reviewer #3: Yes

3. Is the methodology feasible and described in sufficient detail to allow the work to be replicable?

Reviewer #1: Yes

Reviewer #2: Yes

Reviewer #3: Yes

4. Have the authors described where all data underlying the findings will be made available when the study is complete?

Reviewer #1: Yes

Reviewer #2: Yes

Reviewer #3: Yes

5. Is the manuscript presented in an intelligible fashion and written in standard English?

Reviewer #1: Yes

Reviewer #2: Yes

Reviewer #3: Yes

6. Review Comments to the Author

You may also provide optional suggestions and comments to authors that they might find helpful in planning their study.

Reviewer #1: This topic is highly important. Pain scales that are clear, objective and have been validated are hard to find. There is a lot of variation in scales. For example, you can find more than one lameness scales. Thus, ensuring that developed scales abide by certain requirements is important.

This paper was well written and easy to understand. I only have a few comments that should be considered.

Line 59: It is important to consider that pain management practices can be labor intensive. Thus, whatever pain management practices are used must consider labor. As we know, labor is hard to recruit and retain in agriculture.

Line 63: It is also important to consider that using pain scales in production systems that use group housing is very difficult. Identifying compromised animals in a timely fashion is difficult given that the number of employees to the number of animals is very disproportionate and thus, time consuming. Additionally, the high turnover of ag employees makes it even more difficult to identify these animals as employees must be trained. While pain scales work great in controlled settings, in commercial settings they don’t work all that great- due to the time constraints and number of animals per pen. Please ensure that this is mentioned in the paper. What do you propose to make pain identification in commercial setting possible and not labor intensive. Although my comments are not the aim of your paper per se, it is important to address what currently prevents people from using pain scales.

Reviewer #2: Abstract

Lines 35-36:

Suggest exclusion criteria to be stated as other ‘non-ordinal’ pain assessment variables

Introduction

It would be useful for the introduction to outline what COSMIN is.

Lines 53-54:

Suggest altering sentence to something like ‘Society has been increasingly concerned about the impact of pain on farm animal welfare’.

Line 54:

Suggest ‘Farm animals..’ instead of ‘These animals..’

Lines 55-56:

Suggest ‘..do not feel as much pain as small animals..’ if that is what you are stating

Is there a more recent reference for misconception that livestock do not feel pain the same as small animals?

Line 57:

Suggest ‘..lack of knowledge about assessing pain in farm animal species..’

Lines 59-60:

Suggest ‘Pain can cause suffering, fear and stress, negatively impacting animal welfare and sometimes decreasing productivity.’

Line 62:

Suggest removing ‘..and its treatment is a mandatory part of the daily routine of veterinarians.’

Lines 63-64:

Why have you specified behaviour and facial expressions only? There are many other methods used for evaluating pain in farm animals

Lines 64-68:

Why is it ideal to assess pain using scoring systems? There are other methods for assessing pain that I wouldn’t define as scoring systems but that are still valid assessment tools (eg measurement of physiological biomarkers). Perhaps a definition of ‘scoring systems’ as relevant to this review is needed at the beginning of this paragraph. Are you just focusing on behaviour? This needs to me made clear early on.

Lines 63-87:

I think this paragraph needs reworking. It doesn’t flow logically and it is confusing to the reader as to what exactly is of focus for the review.

Materials and Methods

Lines 121-122:

This sentence doesn’t make sense.

Line 142:

What instruments are you referring to? The pain scoring systems?

Line 155:

Delete word ‘articles’.

Lines 157 – 158:

Why was nociceptive testing not included when it could be considered a pain scoring system? Again, I think your definition of ‘pain scoring systems’ needs clarifying earlier in the introduction and/or Materials and Methods.

Lines 200-204:

The structure of the descriptions in brackets is confusing. Suggest amending.

Line 207:

“..findings..”

Line 211:

Suggest removing word ‘recognition’ as this is included as part of pain assessment.

Lines 213:

“..gaps in knowledge..”

Lines 214-215:

Suggest replacing “..with a positive impact in animal welfare” to “..to improve animal welfare”.

Reviewer #3: This report protocol describes the plan for a study of pain scoring systems in farm animals - based on the COSMIN checklist. There are for sure, a need for an overview of the available methodology in terms of pain scoring systems to be used to evaluated pain in farm animals. Hence, this work is needed. I am, however, not really sure that there are enough sources to allow the planned comparisons. Below, I list minor/more specific comments:

Throughout the report protocol, the authors are using 'measurement properties' to describe 'quality of methodology'. I am not sure this term is the best one, and not sure people will understand it either. Perhaps consider changing? Perhaps 'methodological qualities'?

Financial disclosure: How come the authors write ' no specific funding' and later in the acknowledgements writes that the work was part of a research project funded from Brazil. How can both be the truth?

L25: When you write concern - do you mean scientific, public, political, societal - or?

L26: I know what you mean by 'still', but I would recommend to write ... is somewhat neglected ... as an alternative

L26-27: This sentence is not clear. I think I know what you mean, but will the reader understand 'frequently receive less analgesics than...'? Please explain a bit further.

L36: I know that 'pain behaviour' it is used in sci writing, but I will advise not to, as it is what some people consider a weasel word, because many of the behaviours potentially indicating pain are not specific to pain, and by calling it 'pain behaviour' the author may give the reader the impression that we know that these behaviours indicate pain - i.e. it becomes kind of self-fulfilling. Check for example the writing by Nicole Nelson.

L46: How will this classification be done? Based on?

L49: It is not clear to me whether the review and the analysis of the results will be done within or across species?

L53: I like the active language, but who is 'our'?

L56: Here the word livestock is used. Why not farm animals, as was used until now. Be consistent.

L59: I suggest to insert 'may' between pain and cause.

L62: Please provide a reference

L64: I don't understand the choice of the word 'ideally'. Why is this ideally? Couldn't other tools be used?

L68: Please insert a reference

L81: I would replace 'they' with 'these tools' in order to avoid misunderstandings

L92-102: Are more references needed? One for each point, I would suggest?

L94: What do you mean by 'items'

L100: Should 'for example' be inserted after 'i.e. pain)'?

L138: In the introduction, alpacas are mentioned. Why not here?

L153: I would delete the word study, as it is redundant.

L154: See my earlier comment to 'pain behaviours'

L155: Why reviews in plural - why not review?

L177: Is pediatric a term often used when writing about animals?

L122: How will you deal with the lack of publication period or language restriction?

L199: I asked also before - how is this categorization done?

L200-205: Is this described in enough detail?

L211: I suggest to insert 'the management' before 'animal welfare'

7. PLOS authors have the option to publish the peer review history of their article (what does this mean?). If published, this will include your full peer review and any attached files.

Reviewer #1: No

Reviewer #2: No

Reviewer #3: No

---

## [Author Response · Author response to Decision Letter 0]

26 Feb 2021

Additional Editor Comments:

This manuscript is interesting; however there is a major flaw in the interpretation of the data, methodology, improving the clarity of arguments, English language and redaction style. It is necessary to improve the manuscript by examining the questions that need to be clarified in a way. Please check your financial disclosure. For your guidance, you can check the reviewers' comments. Thank you for giving us the opportunity to consider your work.

Answer: Thank you for your comments. However, the authors are uncertain what the editor means by ‘interpretation of the data’. For clarification, this submission pertains to the protocol of the systematic review. Data collection will only be performed once this protocol is approved/accepted for publication. 

With regards to the English language, does the editor refer to the use of Oxford commas or specifically to writing style? 

The questions related to funding have been clarified in the Acknowledgments (i.e. The study received no funding. The PhD received a scholarship).

Thank you again for your consideration.

Reviewer #1: This topic is highly important. Pain scales that are clear, objective and have been validated are hard to find. There is a lot of variation in scales. For example, you can find more than one lameness scales. Thus, ensuring that developed scales abide by certain requirements is important.

This paper was well written and easy to understand. I only have a few comments that should be considered.

Answer: Thank you very much for your review and comments.

Line 59: It is important to consider that pain management practices can be labor intensive. Thus, whatever pain management practices are used must consider labor. As we know, labor is hard to recruit and retain in agriculture.

Answer: Thank you. This is a good point. Considering this reviewer’s observations, the authors have added the questions/items below to the data extraction table. Such information will be collected from each study. With regards to the word ‘labor’ we will attempt to clarify if studies report who is the end-user of such instruments (veterinarians, nurses, farmers and/or general staff looking after animals). 

- Who is the end-user of the instrument? 

- Is training required for the use of the scale?

- Has feasibility been evaluated/reported by the authors in any way? 

- Have authored reported the amount of time that it takes for the instrument to be completed?

Line 63: It is also important to consider that using pain scales in production systems that use group housing is very difficult. Identifying compromised animals in a timely fashion is difficult given that the number of employees to the number of animals is very disproportionate and thus, time consuming. Additionally, the high turnover of ag employees makes it even more difficult to identify these animals as employees must be trained. While pain scales work great in controlled settings, in commercial settings they don’t work all that great- due to the time constraints and number of animals per pen. Please ensure that this is mentioned in the paper. What do you propose to make pain identification in commercial setting possible and not labor intensive. Although my comments are not the aim of your paper per se, it is important to address what currently prevents people from using pain scales.

Answer: Thank you. Indeed, these questions are not the primary goal of the study but are certainly valid if such instruments could eventually be used in commercial farms. These issues will be discussed in the manuscript once results of the study are available. At this time, before data collection, it is our impression that pain scales are just starting to be developed and validated and are still very much limited to controlled environments. However, this is the first step to clearly identify and score species-specific pain-related behaviors.

Answering to your question, perhaps non-labor-intensive pain identification in commercial settings will be based on automated systems in the future (e.g. camera systems that can identify pain-related behaviors). For the success of the latter, pain scales need to be robust and scientifically valid. Thus, this systematic review is trying to answer the question on whether these instruments are scientifically valid and reliable. We do not believe we will be able to answer whether these instruments could be applicable in commercial setting. Nevertheless, the authors will strive to collect as much information as possible from reported studies regarding the issue so that we can at least report the current status of these instruments (i.e. whether the potential for use of such instruments in commercial farms is a possibility or whether research on this topic is in such infancy that extrapolation to commercial farms is nowhere near). Based on these comments, further questions/items have been added to the data extraction table.

- In which setting was the scale developed and validated (experimental farm, commercial farm, hospital)?

- Were animals’ group-housed or individually-housed during pain evaluation?

- How experienced were the evaluators included in the study?

- Do the authors report or discuss the applicability of the instrument in commercial farms? 

- Do the authors report the specific use of the scale? For example: to identify painful animals within a pen; to evaluate pain after surgery; to evaluate pain in hospitalized animals, etc. 

Reviewer #2: Abstract

Lines 35-36:

Suggest exclusion criteria to be stated as other ‘non-ordinal’ pain assessment variables

Answer: Thank you very much for your comments and suggestions. The following has been added to the manuscript: “studies reporting non-ordinal pain assessment variables” (lines 183-184).

Introduction

It would be useful for the introduction to outline what COSMIN is.

Answer: Addition information about COSMIN has been added. However, we believe that an explanation about COSMIN is better placed in the Material and Methods section, rather than in the introduction (lines 216-229). 

Lines 53-54:

Suggest altering sentence to something like ‘Society has been increasingly concerned about the impact of pain on farm animal welfare’.

Answer: Corrected as suggested (lines 56-57). This same sentence is now used in the abstract. 

Line 54:

Suggest ‘Farm animals..’ instead of ‘These animals..’

Answer: Corrected (line 57).

Lines 55-56:

Suggest ‘..do not feel as much pain as small animals..’ if that is what you are stating

Is there a more recent reference for misconception that livestock do not feel pain the same as small animals?

Answer: The sentence has been reworded (lines 57-588). We could not find a more recent study to reference this information. If a new reference becomes available during the systematic review, we will make sure to update this reference on the publication of the results.

Line 57:

Suggest ‘..lack of knowledge about assessing pain in farm animal species..’

Answer: Corrected (lines 61-62).

Lines 59-60:

Suggest ‘Pain can cause suffering, fear and stress, negatively impacting animal welfare and sometimes decreasing productivity.’

Answer: Corrected (lines 63-65).

Line 62:

Suggest removing ‘..and its treatment is a mandatory part of the daily routine of veterinarians.’

Answer: Removed.

Lines 63-64:

Why have you specified behaviour and facial expressions only? There are many other methods used for evaluating pain in farm animals

Answer: This has been clarified in the text (lines 68-85).

Lines 64-68:

Why is it ideal to assess pain using scoring systems? There are other methods for assessing pain that I wouldn’t define as scoring systems but that are still valid assessment tools (eg measurement of physiological biomarkers). Perhaps a definition of ‘scoring systems’ as relevant to this review is needed at the beginning of this paragraph. Are you just focusing on behaviour? This needs to me made clear early on.

Answer: We have changed the term ‘pain scoring systems’ to ‘pain scoring instruments’ throughout the manuscript. Further information has been added to the text to define pain scoring instruments and that we are only referring to measures of pain-related behaviors (lines68-85).

These instruments are generally practical and inexpensive. Other means of pain assessment require specialized equipment and training or are non-specific to pain. This information has been added to the text as requested (lines 68-85).

Lines 63-87:

I think this paragraph needs reworking. It doesn’t flow logically and it is confusing to the reader as to what exactly is of focus for the review.

Answer: This paragraph has been re-written for clarification. 

Materials and Methods

Lines 121-122:

This sentence doesn’t make sense.

Answer: This sentence has been re-written (lines 149-151). 

Line 142:

What instruments are you referring to? The pain scoring systems?

Answer: Yes, it is referring to the pain scoring instruments.

Line 155:

Delete word ‘articles’.

Answer: Deleted (line 185).

Lines 157 – 158:

Why was nociceptive testing not included when it could be considered a pain scoring system? Again, I think your definition of ‘pain scoring systems’ needs clarifying earlier in the introduction and/or Materials and Methods. 

Answer: It was not included because our aim is to evaluate pain scoring instruments based on pain-related behavior or facial expression evaluations. This paragraph has been rewritten. 

Lines 200-204:

The structure of the descriptions in brackets is confusing. Suggest amending. 

Answer: Amended.

Line 207:

“..findings..” 

Answer: Corrected (line 248).

Line 211:

Suggest removing word ‘recognition’ as this is included as part of pain assessment. Answer: Deleted ‘recognition’ from the text (line 252).

Lines 213:

“..gaps in knowledge..” 

Answer: Corrected (line 250).

Lines 214-215:

Suggest replacing “..with a positive impact in animal welfare” to “..to improve animal welfare”. 

Answer: Replaced (lines 254-255).

Reviewer #3: This report protocol describes the plan for a study of pain scoring systems in farm animals - based on the COSMIN checklist. There are for sure, a need for an overview of the available methodology in terms of pain scoring systems to be used to evaluated pain in farm animals. Hence, this work is needed. I am, however, not really sure that there are enough sources to allow the planned comparisons. Below, I list minor/more specific comments:

Answer: Thank you very much for your review and comments. Your concern is valid and part of the reason why we are performing this study. To understand the current literature and help guide future research.

Throughout the report protocol, the authors are using 'measurement properties' to describe 'quality of methodology'. I am not sure this term is the best one, and not sure people will understand it either. Perhaps consider changing? Perhaps 'methodological qualities'?

Answer: Thank you for your concern, but ‘Measurement properties’ is a term from the COSMIN terminology and taxonomy (Mokkink et al. 2010 - The COSMIN study reached international consensus on taxonomy, terminology, and definitions of measurement properties for health-related patient-reported outcomes. J Clin Epidemiol. 63(7):737-45.). We have used this terminology throughout the manuscript to be consistent with COSMIN. Further information has been added for clarification (lines 106-113). 

Financial disclosure: How come the authors write ' no specific funding' and later in the acknowledgements writes that the work was part of a research project funded from Brazil. How can both be the truth?

Answer: The study received no funding. However, the first author (a PhD student) received a scholarship from a funding agency from Brazil. This has been clarified in the acknowledgements. 

L25: When you write concern - do you mean scientific, public, political, societal - or?

Answer: Societal concern. This has been clarified (line 26).

L26: I know what you mean by 'still', but I would recommend to write ... is somewhat neglected ... as an alternative

Answer: This sentence has been removed due to word count.

L26-27: This sentence is not clear. I think I know what you mean, but will the reader understand 'frequently receive less analgesics than...'? Please explain a bit further.

Answer: This has been re-written in the introduction (lines 57-58). However, it had to be removed from the abstract due to word count. 

L36: I know that 'pain behaviour' it is used in sci writing, but I will advise not to, as it is what some people consider a weasel word, because many of the behaviours potentially indicating pain are not specific to pain, and by calling it 'pain behaviour' the author may give the reader the impression that we know that these behaviours indicate pain - i.e. it becomes kind of self-fulfilling. Check for example the writing by Nicole Nelson.

Answer: Changed for “pain-related behaviors” through all text. 

L46: How will this classification be done? Based on?

Answer: This is explained in the materials and methods section. Assessment of quality of included studies will be done based on the COSMIN checklist.

L49: It is not clear to me whether the review and the analysis of the results will be done within or across species?

Answer: This is presented on the ‘Assessment of the measurement properties’ (lines 210-246). Initially, each study will be assessed for ‘methodological quality’ and ‘quality of the findings’ (according to the COSMIN methodology). Thereafter, each instrument will be assessed for ‘quality of evidence’ while considering all the studies available for that instrument which is species-specific. Quality of evidence will be classified as high, moderate, low, or very low. Thus, the final outcome will be the quality of evidence available in the literature for each pain scoring instrument. 

L53: I like the active language, but who is 'our'?

Answer: Reworded (line 56).

L56: Here the word livestock is used. Why not farm animals, as was used until now. Be consistent.

Answer: Corrected (line 57).

L59: I suggest to insert 'may' between pain and cause.

Answer: This sentence was reworded as suggested by another reviewer. 

L62: Please provide a reference

Answer: This sentence has been deleted as requested by another reviewer.

L64: I don't understand the choice of the word 'ideally'. Why is this ideally? Couldn't other tools be used?

Answer: The word ‘Ideally’ was removed, and this paragraph has been re-written as per this and other reviewers’ suggestions. 

L68: Please insert a reference

Answer: Inserted: Luna SPL, de Araújo AL, da Nóbrega Neto PI, Brondani JT, de Oliveira FA, Azerêdo LM dos S, et al. Validation of the UNESP-Botucatu pig composite acute pain scale (UPAPS). PLoS One. 2020;15: e0233552. 

L81: I would replace 'they' with 'these tools' in order to avoid misunderstandings

Answer: Corrected (line 99).

L92-102: Are more references needed? One for each point, I would suggest?

Answer: The references were provided for each point (lines 115-130).

L94: What do you mean by 'items'

Answer: The items or questions included in the scoring instruments.

L100: Should 'for example' be inserted after 'i.e. pain)'?

Answer: This entire paragraph has been re-written as requested by another reviewer. 

L138: In the introduction, alpacas are mentioned. Why not here?

Answer: Thank you. This has been corrected to ‘camels’. The word ‘alpacas’ was kept by mistake (line 105). 

L153: I would delete the word study, as it is redundant.

Answer: Deleted (line 182)

L154: See my earlier comment to 'pain behaviours'

Answer: Changed to pain-related behaviors (line 183).

L155: Why reviews in plural - why not review?

Answer: Corrected (line 184).

L177: Is pediatric a term often used when writing about animals?

Answer: Yes, this terminology is commonly used to refer to the different life stages of animals. 

L122: How will you deal with the lack of publication period or language restriction?

Answer: These were chosen to increase our chances of finding relevant articles. Within our research group we speak 7 languages and would be able to evaluate articles in these languages if they come up. If articles in other languages appear, we will look for translators to help us with those. 

L199: I asked also before - how is this categorization done?

Answer: Please see our response above.

L200-205: Is this described in enough detail?

Answer: We believe so. We welcome any suggestions for modification. 

L211: I suggest to insert 'the management' before 'animal welfare'

Answer: This sentence was rewritten to “potential target for future studies in farm animals to improve animal welfare” as a requested by another reviewer (lines 251-252)

Note to all reviewers: The section “Protocol registration” was removed. We have been in contact with SyRF and they reported that they will no longer register systematic reviews from veterinary medicine. Considering that the present submission already involves the publication of the protocol before data collection, registration of the protocol would be redundant.

---

## [Decision Letter · Decision Letter 1]

20 Apr 2021

PONE-D-20-37059R1

Measurement properties of pain scoring instruments in farm animals: a systematic review protocol using the COSMIN checklist

PLOS ONE

Dear Dr. Steagall,

Thank you for submitting your manuscript to PLOS ONE. After careful consideration, we feel that it has merit but does not fully meet PLOS ONE’s publication criteria as it currently stands. Therefore, we invite you to submit a revised version of the manuscript that addresses the points raised during the review process.

Please make minor revision in the revised MS />==============================

We look forward to receiving your revised manuscript.

Kind regards,

Arda Yildirim, Ph.D.

Academic Editor

PLOS ONE

Journal Requirements:

Additional Editor Comments (if provided):

For your guidance, you can check the reviewers' comments. Thank you for giving us the opportunity to consider your work.

Reviewers' comments:

Reviewer's Responses to Questions

**Comments to the Author**

1. Does the manuscript provide a valid rationale for the proposed study, with clearly identified and justified research questions?

Reviewer #1: Yes

Reviewer #2: Yes

Reviewer #3: Yes

2. Is the protocol technically sound and planned in a manner that will lead to a meaningful outcome and allow testing the stated hypotheses?

Reviewer #1: Yes

Reviewer #2: Yes

Reviewer #3: Yes

3. Is the methodology feasible and described in sufficient detail to allow the work to be replicable?

Reviewer #1: Yes

Reviewer #2: Yes

Reviewer #3: Yes

4. Have the authors described where all data underlying the findings will be made available when the study is complete?

Reviewer #1: Yes

Reviewer #2: Yes

Reviewer #3: Yes

5. Is the manuscript presented in an intelligible fashion and written in standard English?

Reviewer #1: Yes

Reviewer #2: Yes

Reviewer #3: Yes

6. Review Comments to the Author

You may also provide optional suggestions and comments to authors that they might find helpful in planning their study.

Reviewer #1: This is a well written paper. The topic is of upmost importance. Validated scales are difficult to find and use. I think this manuscript provides information that will be very helpful in the development of such scales.

Reviewer #2: This manuscript is greatly improved following suggested reviewers' comments and my recommendation is to accept in current form.

Reviewer #3: Dear authors

Thank you for the revised manuscript. I have only one comment left: I notice that 'pain behaviours' are now changed to 'pain-related behaviours'.

In the first version of the manuscript, I mentioned that I know that 'pain behaviour' it is used in sci writing, but I advised you to not do that, as it is what some people consider a weasel word, because many of the behaviours potentially indicating pain are not specific to pain, and by calling it 'pain behaviour' the author may give the reader the impression that we know that these behaviours indicate pain - i.e. it becomes kind of self-fulfilling. Check for example the writing by Nicole Nelson.

The re-wording from 'pain behaviour' to 'pain-related' behaviors is a move in the right direction, but I still consider it problematic. I would write 'behaviours potentially indicating pain' or something like that.

7. PLOS authors have the option to publish the peer review history of their article (what does this mean?). If published, this will include your full peer review and any attached files.

Reviewer #1: **Yes: **Arlene Garcia

Reviewer #2: No

Reviewer #3: No

---

## [Author Response · Author response to Decision Letter 1]

20 Apr 2021

Manuscript PONE-D-20-37059R1

Authors: Thank you very much for your comments.

Reviewer #1: This is a well written paper. The topic is of upmost importance. Validated scales are difficult to find and use. I think this manuscript provides information that will be very helpful in the development of such scales.

Answer: Thank you for your thoughtful review.

Reviewer #2: This manuscript is greatly improved following suggested reviewers' comments and my recommendation is to accept in current form.

Answer: Thank you for your thoughtful review.

Reviewer #3: Dear authors

Thank you for the revised manuscript. I have only one comment left: I notice that 'pain behaviours' are now changed to 'pain-related behaviours'.

In the first version of the manuscript, I mentioned that I know that 'pain behaviour' it is used in sci writing, but I advised you to not do that, as it is what some people consider a weasel word, because many of the behaviours potentially indicating pain are not specific to pain, and by calling it 'pain behaviour' the author may give the reader the impression that we know that these behaviours indicate pain - i.e. it becomes kind of self-fulfilling. Check for example the writing by Nicole Nelson.

The re-wording from 'pain behaviour' to 'pain-related' behaviors is a move in the right direction, but I still consider it problematic. I would write 'behaviours potentially indicating pain' or something like that.

Answer: Thank you again for your comments. We have now replaced “pain-related behaviors” to the wording (or similar) suggested by the reviewer throughout the manuscript.

---

## [Editor Report · Decision Letter 2]

27 Apr 2021

Measurement properties of pain scoring instruments in farm animals: a systematic review protocol using the COSMIN checklist

PONE-D-20-37059R2

Dear Dr. Steagall,

We’re pleased to inform you that your manuscript has been judged scientifically suitable for publication and will be formally accepted for publication once it meets all outstanding technical requirements.

Kind regards,

Arda Yildirim, Ph.D.

Academic Editor

PLOS ONE

Additional Editor Comments (optional):

Thanks for your hard work.
---

## [Editor Report · Acceptance letter]

7 May 2021

PONE-D-20-37059R2 

Measurement properties of pain scoring instruments in farm animals: a systematic review protocol using the COSMIN checklist 

Dear Dr. Steagall:

I'm pleased to inform you that your manuscript has been deemed suitable for publication in PLOS ONE. Congratulations! Your manuscript is now with our production department. 

Kind regards, 

on behalf of

Prof. Dr. Arda Yildirim 

Academic Editor

PLOS ONE